# Characterization of Dysphagia Thickeners Using Texture Analysis—What Information Can Be Useful?

**DOI:** 10.3390/gels8070430

**Published:** 2022-07-09

**Authors:** Raquel Baixauli, Mireia Bolivar-Prados, Kovan Ismael-Mohammed, Pere Clavé, Amparo Tárrega, Laura Laguna

**Affiliations:** 1Institute of Agrochemistry and Food Technology (IATA, CSIC), 46980 Valencia, Spain; rbaixauli@iata.csic.es (R.B.); atarrega@iata.csic.es (A.T.); 2Gastrointestinal Physiology Laboratory, Hospital de Mataró, Universitat Autònoma de Barcelona, 08304 Mataró, Spain; mbolivar@csdm.cat (M.B.-P.); kovan.ismael@yahoo.com (K.I.-M.); pere.clave@ciberehd.org (P.C.); 3Centro de Investigación Biomédica en Red de Enfermedades Hepáticas y Digestivas (Ciberehd), 08304 Barcelona, Spain

**Keywords:** texture analyzer, thickener, dysphagia

## Abstract

Besides shear viscosity, other texture parameters (adhesiveness or cohesiveness) might be relevant for safe swallowing in people suffering from oropharyngeal dysphagia. Shear viscosity is assessed through protocols developed using a viscometer or a rheometer. In contrast, protocols and instruments (capillary break-up rheometer) to assess adhesiveness and cohesiveness are less common and much less developed. Other equipment such as texture analyzers can provide useful information on food properties. Here, we aimed to explore different texture analyzer settings (type of test, probe, and protocol) to characterize four commercial dysphagia thickeners at the shear viscosity levels recommended by manufacturers. Among the tests used (extrusion or penetration) with the different probes (disc, cone and shape holder, sphere, or cylinder), cone extrusion provided information about adhesivity, disc extrusion about sample cohesiveness, and sphere about penetration and sample elasticity. The test speeds used influenced the results, but only one speed is needed as the different speeds provided the same fluid information; for easiness, it is proposed to use 1 mm/s. Comparing the texture analyzer results with viscosity values obtained at different shears, the texture analyzer parameters reflected information that differ from shear viscosity. This information could be relevant for the therapeutic effect of thickening products and food characterization.

## 1. Introduction

For patients with oropharyngeal dysphagia (OD), the viscosity of the fluids they consume must be adapted by adding commercial thickener (starch, modified starch, xanthan gum, tara gum, guar gum, and maltodextrins) to increase viscosity, as shear viscosity is the main factor linked to safe swallowing [1,2]. These thickeners comprise starch and/or gums and thicken the sample without need of heating and within a short time [3].

The correct dose of thickener to be prescribed to a patient can be assessed by clinical trials (testing with patients). To diagnose OD, a Video fluoroscopy (VFS) is performed; briefly, the patient is seated in a lateral projection, and different viscosities and volumes are given by a safety algorithm [4]. This technique allows determining safety and efficacy impairments including their clinical dysphagia signs such as dysphonia, dysarthria, abnormal volitional cough, abnormal gag reflex, cough after swallowing, and voice change after swallowing [5,6]. To perform an optimal clinical procedure, viscosities must be known at the shear rate of 50 s^−1^ (estimated value in the oral stage) and 300 s^−1^ (estimated value at the pharyngeal phase) using a viscometer or a rheometer [7]. Then, through VFS, observing the swallowing movements while drinking the thickened products determines whether the patient suffers from swallowing impairments and the safest and optimal viscosity level to be prescribed. Shear viscosity has been identified as the main parameter linked to the therapeutic effect (safe swallow prevalence) of thickening products [1] in contrast with extensional viscosity [8].

To help dysphagia sufferers, food science research has traditionally focused on the study and modification of food and liquids’ rheological properties (those that govern food viscoelasticity and flow), when adding different thickeners to create more viscous or consistent pureed food and liquids [9,10,11,12,13,14]. Lately, research into dysphagia products have also placed their attention on the characterization of this diet beyond the traditional shear viscosity measurement. In this line, measurement of the extensional flow has been shown to provide further information [15,16] of oropharyngeal residues. Moreover, the knowledge of tribological properties can provide an understanding of the thickener’s surface properties [17] and the interaction of dysphagia thickeners with the oral surface that will provide stickiness or slippery characteristics to the thickeners [18]. Further, according to [19,20], other parameters might also apply to patients with OD, such as adhesiveness or springiness. For example, the measurement of adhesiveness can be important in patients with OD; sticky foods such as mochi (Japanese rice cake) have been associated with choking, as they also stick to oropharyngeal structures blocking air pathways [20]. These parameters cannot be obtained by a viscometer or a rheometer, but they could be obtained using empirical tests performed by the texture analyzer.

However, in dysphagia thickeners, using the texture analyzer has been scarcely studied. For dysphagia thickeners, the forward extrusion test has been used to study the influence of time and temperature at different thickener concentrations (nectar, honey, and pudding level) [21]. Authors placed the thickened drink in a cylindric glass whilst extruding by a flat disc probe and obtained different parameters after studying the curves such as firmness, adhesiveness, and work that differentiated the samples [21]. Cevoli et al. [22], also using a back-extrusion test, analyzed different hydrocolloids not specific for dysphagia use (carboxymethylcellulose, tara gum, guar gum, locust bean gum, xanthan gum, and sodium alginate), which could also indicate significant differences between samples. In food matrices of viscosities similar to dysphagia thickeners, the texture analyzer has been a good characterization tool using extrusion or penetration tests. Agudelo et al. [23] used an extrusion test to study the maximum force needed to extrude fruit fillings for pastries, whilst other works [24] used the extrusion test to characterize yogurts with different fat level firmness or to study the effect of different fibers’ concentrations on frozen/thawed potatoes purees [25].

Therefore, a texture analyzer has the potential to characterize further properties of dysphagia thickeners. However, one must define the protocols and key dysphagia parameters to be obtained using a texture analyzer. Furthermore, investigating which texture analyzer tests, probes, and experimental settings are best is required as they can be a quick, understandable tool to use in a medical environment.

In summary, beyond viscosity, there are other mechanical properties of dysphagia thickeners that might help in the patient’s safety and, so far, there is no objective measurement to quantify them. For that, there is a need to explore which texture analyzer tests and specific settings could provide more information on dysphagia thickeners’ mechanical properties and if this information complements or overlaps that offered by the information obtained when measuring the shear viscosity.

This study has three objectives. The first one is to explore the information that different texture analyzer tests (type of test, probe, and settings) can provide for dysphagia thickeners’ characterization considering the viscosity levels described in the user instructions of commercial thickeners (nectar and pudding consistencies). The second one is to elucidate if the information provided by the texture analyzer complements or overlaps the information provided by the shear viscosity. Finally, the third objective is to provide a guide with the optimum protocol test to characterize dysphagia thickener.

## 2. Materials and Methods

### 2.1. Materials

Four commercial thickeners were used (A—ThickenUp, Nestlé España S.A., Esplugues de Llobregat, Spain; B—NutAvant, Persan Farma S.L., Las Palmas de Gran Canaria, Spain; C—FontActiv, Laboratoires Grand Fontaine S.L., Barcelona, Spain; D—Nutilis Powder, Nutricia S.R.L., Fulda, Germany). Selected thickeners varied in composition and represented the common ingredients used by dysphagia patients (see Table 1).

### 2.2. Preparation of the Thickened Drinks

Thickened drinks were prepared in water at room temperature. The matrix in which drinks are dissolved and the temperature are important, as both factors have been shown to affect the final viscosity. For instance, in a previous work, it was elucidated that drinks with soluble solids (for example, milk and orange juice) had more viscosity than drinks without soluble solids (e.g., coffee) [26]. In this study, water was selected as it is the most used drink to calm thirstiness.

The quantity of thickener used was as described by each manufacturer’s instructions for the level of nectar and pudding, as shown in Table 2.

Each thickener was dissolved in 100 mL of distilled water using an overhead paddle stirrer at a speed of 135 rpm for 1 min for nectar samples and 2 min for pudding samples (25 °C). Samples rested for 2 min before measurement.

### 2.3. Texture Analyzer Tests

The texture measurements were conducted with a TA-XTPlus Texture Analyzer (Stable Microsystems, Goldalming, UK) equipped with Texture Exponent software (version 2.0.7.0. Stable Microsystems), with a 30-kg load cell and 0.1 N of trigger force. The texture analyzer captures the force, distance, and time needed to—depending on the probe—attach, extrude, penetrate, or compress the dysphagia thickener whilst the arm of the texture analyzer moves down and returns to its initial position. Data are registered once the trigger force is achieved. These data are also shown graphically whilst the test is performed.

Four tests frequently used in the evaluation of textural attributes of semi-solid food products were performed: two back-extrusion tests and two penetration tests [27]. The tests are extensively described in Section 2.3.1 and Section 2.3.2.

All tests were conducted at 1, 3, 5, and 10 mm/s and at 25 °C. This was the room temperature, as no device to maintain the temperature in the texture analyzer was available. The selection of the temperature is important as the physical properties depend on it; for example, viscosity, according to the Arrhenius equation, depends on temperature. For the four tests, the pretest and the post-test speeds were 10 mm/s. Table 3 summarizes the tests and parameters. Three replicates were performed per sample.

#### 2.3.1. Extrusion Tests

##### Extrusion with Disc

A back-extrusion test was conducted using a holding sample container of 50 mm-diameter and 75 mm-height, and a disc plunger of 35 mm-diameter (back-extrusion disc A/BE35). The distance the disc plunger traveled was 30 mm. From the force–displacement curves, values for the gradient, maximum positive force, positive area, final positive force, and maximum negative force were obtained.

##### Extrusion with Cone and Cone Shape Holder

The texturometer was fitted with a TTC Spreadability Rig containing a male cone (90° and 40 mm diameter) that matched a glass containing the female cone fixed in an HDP/90 platform. The sample was placed in the female cone. The force when penetrating until a depth of 22.5 mm was recorded. From the force–displacement curves, the maximum peak force and the positive area during the downstroke, and the maximum negative force and the adhesiveness during the upstroke were obtained.

#### 2.3.2. Penetration Test

##### Penetration with a Spherical Probe

A spherical probe (P/1S) (diameter 25.4 mm) was used to penetrate up to a distance of 20 mm, using a trigger force of 0.1 N. Samples were filled to the top of plastic cups (50 mm-diameter, 40 mm-height). The parameters obtained were the maximum peak force and positive area under the curve.

##### Penetration with a Cylindrical Probe

In this test, the thickeners were penetrated with a cylindrical probe (diameter 25 mm) (P/25P) up to a distance of 20 mm using a trigger force of 0.1 N. Samples were filled to the top of plastic cups (50 mm-diameter, 40 mm-height). The parameters obtained were the maximum peak force and positive area under the curve.

### 2.4. Rheological Measurement of Thickeners

Rheological characterization of thickeners (shear viscosity) was performed using a rheometer (AR-G2, TA Instruments, Crawley, England). A parallel rough plate geometry system (Plate SST ST XHatch 40 mm Smart-SW) and Peltier plate steel with an axial gap of 1500 μm were used for all measurements. The thickener flow curves were obtained as a function of increasing the shear rate from 1 to 350 s^−1^ at 37 °C. The values of viscosity as a function of shear rate at 10, 50, and 300 s^−1^ were collected.

### 2.5. Statistical Analysis

Analysis of variance was used to study the variability between samples at 1 mm/s and the variability between different speeds for one thickener. Significant differences between individual samples were determined using Fisher’s least significant difference test (*p* = 0.05). The overall variability in the texture parameters with the viscosities obtained at 10, 50, and 300 s^−1^ was analyzed using principal component analysis (PCA). All the calculations were conducted with XLSTAT 2020.4.1 (Addinsoft, Paris, France).

## 3. Results and Discussion

### 3.1. Characterizing Dysphagia Thickeners Using the Texture Analyzer

#### 3.1.1. Extrusion Tests

Figure 1 indicates an example of the curve obtained at the nectar level for the two-extrusion test.

As observed in the force–distance curves, when the samples are extruded by a disc plunger in a cylindric cup, there is an initial slope while the disc passes through the sample, forcing it to flow, until a point when the sample locates itself over the disc. After that maximum point of the slope, there is a plateau until the test is finished. From this curve, the gradient can be obtained to indicate the initial travel of the disc plunger through the thickener matrix—the initial force to flow, as a measurement of initial resistance to extrusion, and the positive area under the curve, as a measure of system consistency. Furthermore, the final force or maximum force reflects thickener firmness [22]. In this case (extrusion by a disc), as emphasized by Cevoli et al. [22] when measuring other hydrocolloids, the negative values of the curves cannot be used as adhesivity; this is because the sample is lifted to the upper surface of the disc while rising, and that value only reflects the weight of the sample and not its adhesivity. Other authors have used this value to assess the easiness of the sample flowing from the disc, which can reflect sample cohesiveness [28].

The values of the disc extrusion curve are presented in Table 4. At the nectar level, gradient values were higher for thickener C than for the rest of the thickeners; however, the initial force, maximum positive force, and final force of thickener D indicated the highest values. This meant that the gradient values differed from the trend of the rest, although visually, the gradient might seem similar among samples. This could mean that the initial traveling for nectar consistency does not reflect thickener matrix information.

At pudding concentration, except for the gradient, thickener D is still the sample with significantly higher values of positive area, initial, and final force.

In the extrusion test with a cone, the cone produces a shear force, squeezing out the fluid radially. As observed in the graph (Figure 1), this is translated as an increment in the force to travel through the sample. The area under this first part of the curve is positive and is related to the firmness or the energy required to deform and, in this case, force the thickener to flow. When the specified penetration (22.5 mm) is reached, the probe withdraws from the sample at the post-test speed; meanwhile, the negative curve appears. This negative area corresponds to the adhesivity that was defined by Bourne [29] as the force needed to remove the product adhered from another body, in this case, from the probe.

Table 4 indicates the values obtained from the extrusion with cone curve. At both consistencies studied, most values obtained (maximum positive force, negative area, and maximum negative force) were higher for thickener D.

Therefore, the extrusion with a disc or cone differentiated, at nectar and pudding levels, differences among thickeners. Both tests indicated that thickener D was significantly more cohesive, adhesive, and firm than thickeners A, B and C.

#### 3.1.2. Penetration Test

The penetration test curves are shown in Figure 2, and the data obtained from the curve are shown in Table 5. Both tests performed using the cylinder or the sphere indicate a very similar profile—a positive curve that keeps increasing—as the drink is being displaced without disruption as no surface is punctured, and a negative curve is drawn as the probe returns to the start position. In previous research, when measuring dysphagia thickeners, it was stated that the compression force made by the penetration plunger of the texture analyzer was measuring the resistance of the drink to an imposed displacement, and this might be closer to forces when drinking a thickener [30].

The penetration test with a sphere is recommended when the sample is extremely thin, as could be the case of the dysphagia thickeners. Here, the final force and area were obtained, and no adhesivity values were recorded because the sample slipped completely from the geometry. For nectar consistency values, penetration with the sphere indicates that less force was required to pass the sphere through thickener B; consequently, the area was also less. For pudding level, sphere penetration force was higher for sample B and the lowest was for sample A. In this study, only one diameter sphere was investigated, and it needs to be considered that changing the diameter can cause the deformation to increase (Bourne, 2002) but the differences among samples will remain similar.

When using the cylinder, due to the small contact area of the probe with the sample, it could not capture the adhesivity. The values from the penetration with cylinder (Table 4) shows that at the nectar consistency level, the highest value for the area was C, and for pudding, the highest was thickener D. Regarding the final maximum pick, at nectar level, the significantly lower force was for thickener A. For the pudding consistency level, using penetration with the cylinder discriminated samples into different groups, with D having the maximum force followed by B, C, and A.

### 3.2. Effect of Test Speed

When the speed is changed, the area and the gradient directly change, as when increasing the speed, the distance traveled increases for the same amount of time. Therefore, this study only focused on knowing how, by applying different speeds, the forces needed to extrude or penetrate the dysphagia thickeners are affected. For each thickener, the forces at different speeds are shown in Table 6 and Table 7.

For both nectar and pudding consistencies, the registered force increases with the test speed. The increment in registered force with an increment in test speed has been reported by other authors [31,32,33] and described with a logarithmic relationship [33]. Similar to the pudding level of dysphagia thickeners, in gelled systems, the force increment with the speed has been attributed to the relaxation time of the specimen [32]. With fluids, as the dysphagia thickeners at nectar level, the thickened liquid is forced to flow faster, resulting in more force needed to move it.

Regarding the selection of the speed to perform the test, the presented results show that the information provided by any speed reflects the same information trend, and one speed would be enough to understand the textural behavior of the samples. For ease of results interpretation, the selection of 1 mm s^−1^ could be used.

For ease of results interpretation, the selection of 1 mm s^−1^ could be used as it can discriminate between the different samples well. It is important to keep the same speed for comparing the samples as many of the textural parameters are based on energy considerations, for example, the area under the force–displacement curve represents energy. Further, when energy parameters are evaluated using the force–time curve, it is necessary to divide the results by the value of the speed of travel of the probe to obtain the distance traveled; therefore, it is important to use the same speed to compare the results [34].

### 3.3. Relationship of Texture Analyzer Parameters with Traditional Viscosity Measurements

For the characterization of dysphagia thickeners, previous studies have proposed to use shears at 10 s^−1^ to link with sensory perceptions [35,36], and 50 s^−1^ or 300 s^−1^ to relate viscosity either with oral behavior or pharyngeal behavior of patients who suffered OD [37]. To consider the suggested shears, in this study, the apparent viscosity values at 10, 50, and 300 s^−1^ of shear rate were obtained from the flow curves for the different thickeners and are presented in Table 8. Comparing the viscosity values at increasing shear rates, a decrease is observed; this is because dysphagia thickeners had a shear-thinning behavior, as described by other authors [8,26]. At nectar level, thickener A at 10 and 50 s^−1^ was the most viscous sample, followed by thickeners B, D and C. For pudding level, thickener A was the least viscous, whereas the other three had higher and similar viscosities for the selected shear rates.

A PCA was applied to understand the relationships between the rheometer and texture analyzer measurement. For the nectar level, the PCA explained 82.64% of total variability, and for the pudding level, 99.2% (Figure 3). The samples were distributed according to the differences and similarities of the different measurements.

For the nectar level, PC1 distributes the samples by the texture analyzer results, separating sample D with the highest values of extrusion forces, penetration forces, and adhesivity values (force and area obtained with extrusion with cone) from the rest. PC2 separates the samples by viscosity; so, B is the sample with the highest viscosity (obtained with the rheometer) and highest consistency (positive area values obtained by disc extrusion).

For the pudding level, PC1 separates samples D, B, and C from A. This is because thickeners D, B, and C had the highest registered forces and viscosities. Furthermore, sample A is separated from the rest because it was the least viscous and the one that offered less resistance during extrusion and penetration with the cylinder. Here, viscosity and most of the texture analyzer results are in separated quadrants, meaning that although they indicated the same trend in sample differences, the information provided was different.

Comparing the different tests conducted with a texture analyzer, extrusion with cone was the only test that could measure a parameter related with adhesivity, which has been mentioned to be important for patients with dysphagia, as adhesive food textures need more lingual effort to form the bolus and propel it down the pharynx [20]. Further, extrusion with disc was the only test that could measure a parameter related with cohesiveness, which has also been mentioned as an important parameter as highly cohesive liquids can cause swallowing difficulty due to their high resistance to deformation during swallowing [31]. Furthermore, extrusion could be closer to the forces that thickeners experiment during oral processing than to the penetration test, such as the tongue force when squeezing against the palate and when propelling the food toward the pharynx, or the esophageal forces when squeezing the bolus to the stomach. Moreover, penetration tests also discriminated among the different thickeners. 

The viscosity results obtained with the rheometer and the results obtained with the texture analyzer complement each other; they provide different information covering the different dimensions or properties that the thickeners have.

This study has some limitations, the first one is the control of temperature in the texture analyzer. In the future, to be able to control the temperature along with the measurement it would be interesting to incorporate a temperature-control device in the texture analyzer such as a thermal cabinet or a Peltier plate. Moreover, it would help to explore other physical properties beyond shear viscosity, such as surface properties or extensional rheology, to further understand the thickeners’ response to the rheometer.

In the academic world, physical food science has been using texture analyzers for a long time to develop and adjust texture analyzer parameters and to characterize new lab-manufactured foods [29]. From a practical point of view, within the food industry, the texture analyzer is used in product development and quality control [38]. In the medical environment, there is a growing interest in using the texture analyzer, precisely to characterize the food that is being given to dysphagia patients [39]; further, the use of the texture analyzer could benefit the hospital environment as a substitute for some of the current measurements used in the pharmacopeia. However, as a limitation of using a texture analyzer, it needs to be cited that its acquisition requires extra funding that could be considered no directly related to the healing aspects of a hospital but useful for research, and it also requires trained personnel to perform the experiments and analyze and interpret the data.

## 4. Conclusions

This study proposed using a texture analyzer to characterize different commercial dysphagia thickeners when dissolved in water. With penetration tests, forces and areas could discriminate thickeners. Extrusion tests provided a higher number of parameters than penetration such as adhesion, firmness, and cohesivity; these are reported as important parameters for patients with dysphagia and cannot be obtained using a rheometer or viscometer. In summary, from the extrusion with disc, it was possible to obtain the consistency, firmness, and cohesivity of the thickeners; in addition to this, by extruding with the cone, it was possible to obtain the adhesivity. 

Changing the test speed affected the results, as a higher speed translated into higher force and area, but the difference registered among the samples had the same trend. Therefore, performing the test at one speed would be satisfactory to characterize dysphagia thickeners; thus, 1 mm s^−1^ could facilitate results interpretation. As there are different phenotypes of patients with dysphagia (elderly, neurodegenerative diseases, poststroke, and head and neck cancer), the selection of this speed should be adapted to the oral motor response of each group to accurately simulate the swallowing abilities of each patient.

This study highlights that the characterization of thickened fluids using a texture analyzer provides other rheological parameters that go beyond shear viscosity. In the future, it would be interesting to perform characterization and classification of drinks and foods for dysphagia users using a texture analyzer to provide meaningful information that can be related to therapeutic effects on safety and efficacy of swallowing in patients with dysphagia.

## Figures and Tables

**Figure 1 gels-08-00430-f001:**
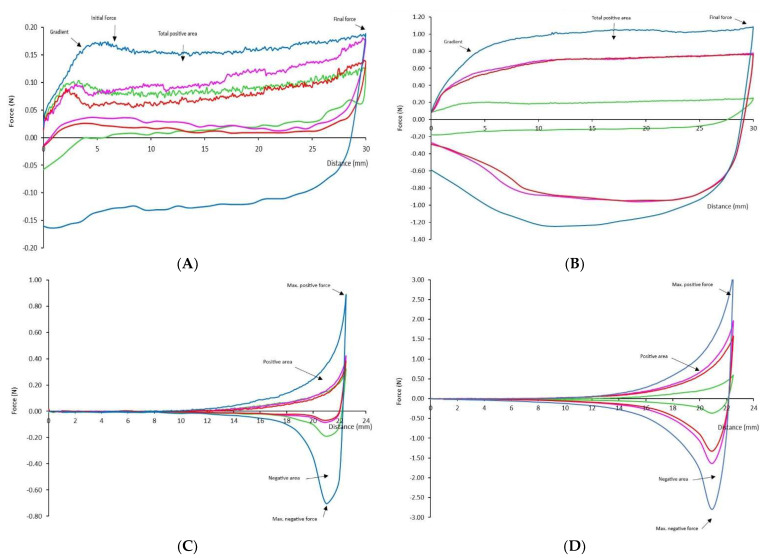
Force–deformation curve of the four thickeners derived from the extrusion test with disc ((**A**): nectar, (**B**): pudding) and cone (**C**): nectar, (**D**): pudding) at 1 mm/s. Different colors indicate different thickeners: green—thickener (**A**), pink—thickener (**B**), red—thickener (**C**), and blue—thickener (**D**).

**Figure 2 gels-08-00430-f002:**
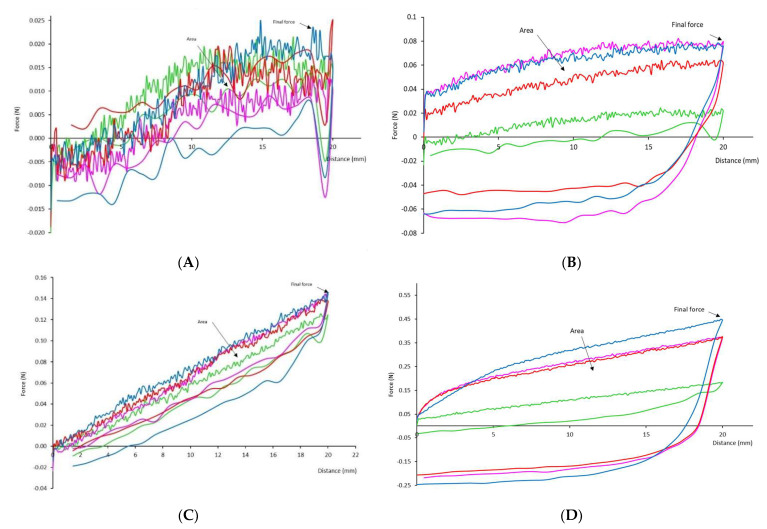
Force–deformation curve of the four thickeners derived from the penetration test with sphere probe (top) and cylinder probe (bottom) at speed of 1 mm s^−1^ (Left: nectar; Right: pudding). Different colors indicated different thickeners: green—thickener (**A**), pink—thickener (**B**), red—thickener (**C**), and blue—thickener (**D**).

**Figure 3 gels-08-00430-f003:**
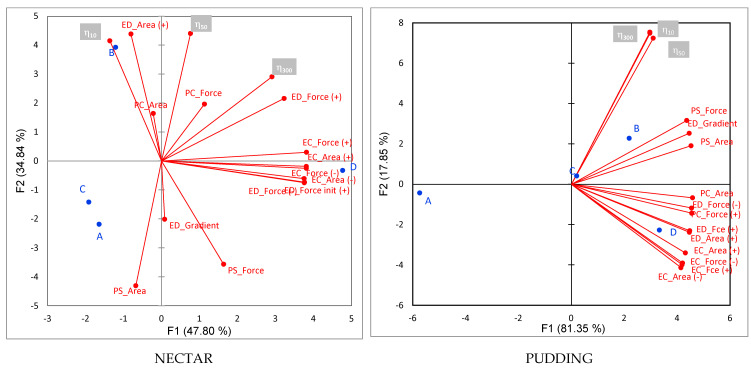
PCA plot of texture and rheological measurement for the four thickeners at two consistency levels: nectar and pudding. EC—extrusion with cone, ED—extrusion with disc, PC—penetration with cylinder, PS—penetration with the sphere. In blue color are the samples and in the grey boxes are the apparent viscosities at different shear rates (10, 50, and 300 s^−1^).(Thickeners: A, B, C, D).

**Table 1 gels-08-00430-t001:** Selected dysphagia thickeners composition.

Thickener	Ingredients
A	Maltodextrin, xanthan gum, and potassium chloride
B	Modified maize starch
C	Modified maize starch
D	Maltodextrin, thickeners (modified starch (maize), tara gum, xanthan gum, guar gum)

**Table 2 gels-08-00430-t002:** Quantity of commercial thickeners used.

Thickeners	Grams per 100 mL	
	Nectar	Pudding
A	1.2 g	3.6 g
B	4.1 g	8.2 g
C	4.0 g	8.0 g
D	5.0 g	9.0 g

**Table 3 gels-08-00430-t003:** Instrumental parameters obtained from the force–distance curves for the different tests.

Test Type	Probe	Selected Parameter	Units
Extrusion with disc	With disc (back-extrusion disc A/BE35)	Gradient	N/mm
		Positive area from the curve	N.mm
		First positive peak force	N
		Final positive peak force	N
		Maximum negative force peak	N
Extrusion with cone	With cone TTC Spreadability Rig	Maximum positive peak force	N
		Positive area from the curve	N.mm
		Maximum negative peak force	N
		Negative area from the curve	N.mm
Sphere penetration		Positive area from the curve	N.mm
		Maximum positive peak force	N
Cylinder penetration		Positive area from the curve	N.mm
		Maximum positive peak force	N

**Table 4 gels-08-00430-t004:** Back-extrusion values with disc or cone at a speed of 1 mm s^−1^ at nectar and pudding concentrations.

		Extrusion with Disc						Extrusion with Cone		
	Thickener	Gradient (N/mm)	Positive Area (N.mm)	Max. Positive Force (N)	Final Positive Force (N)	Max. Negative Force (N)	Max. Positive Force (N)	Positive Area (N.mm)	Max. Negative Force (N)	Negative Area (N.mm)
Nectar level	A	0.028 bc (0.004)	4.742 b (0.147)	0.107 b (0.003)	0.128 d (0.004)	0.087 b (0.002)	0.337 b (0.014)	0.837 b (0.015)	0.193 b (0.006)	0.648 b (0.062)
	B	0.027 c (0.001)	5.657 a (0.239)	0.095 c (0.001)	0.169 b (0.015)	0.069 bc (0.011)	0.424 b (0.038)	0.814 b (0.056)	0.163 b (0.114)	0.418 c (0.096)
	C	0.041 a (0.004)	4.837 b (0.494)	0.093 c (0.009)	0.147 c (0.008)	0.067 c (0.008)	0.364 b (0.022)	0.683 b (0.030)	0.073 b (0.011)	0.295 c (0.041)
	D	0.033 b (0.002)	4.831 b (0.206)	0.174 a (0.009)	0.195 a (0.006)	0.178 a (0.015)	0.856 a (0.085)	1.714 a (0.210)	0.876 a (0.448)	1.868 a (0.144)
Pudding level	A	0.025 c (0.002)	6.010 c (0.121)	----	0.248 c (0.007)	0.177 d (0.000)	0.594 d (0.028)	1.425 d (0.017)	0.353 d (0.016)	0.051 d (0.001)
	B	0.089 a (0.006)	20.112 b (0.945)	-----	0.788 b (0.028)	0.970 b (0.013)	1.891 b (0.057)	4.822 b (0.208)	1.603 b (0.021)	0.139 b (0.006)
	C	0.070 b (0.009)	19.467 b (0.431)	-----	0.759 b (0.016)	0.929 c (0.024)	1.563 c (0.042)	4.032 c (0.108)	1.308 c (0.024)	0.116 c (0.005)
	D	0.079 ab (0.002)	28.216 a (0.617)	-----	1.088 a (0.023)	1.249 a (0.019)	3.138 a (0.132)	7.468 a (0.069)	2.774 a (0.071)	0.228 a (0.006)

Means values in the same column, within the same concentration level (nectar or pudding) with different letters, are significantly different (*p* < 0.05) according to Fisher’s least significant difference. For each mean, standard deviation is in parenthesis.

**Table 5 gels-08-00430-t005:** Penetration test values with sphere or cylinder at a speed of 1 mm s^−1^ at nectar and pudding concentrations.

		Penetration with Sphere		Penetration with Cylinder	
	Thickener	Area (N.mm)	Final Force (N)	Area (N.mm)	Final Force (N)
Nectar level	A	0.326 ab	0.020 a	2.167 c	0.126 b
		(0.052)	(0.003)	(0.043)	(0.005)
	B	0.160 c	0.014 b	2.404 ab	0.140 a
		(0.04)	(0.002)	(0.123)	(0.004)
	C	0.351 a	0.024 a	2.493 a	0.143 a
		(0.051)	(0.002)	(0.031)	(0.005)
	D	0.264 b	0.024 a	2.342 b	0.141 a
		(0.023)	(0.002)	(0.076)	(0.005)
Pudding level	A	0.242 c	0.025 c	2.276 c	0.186 c
		(0.033)	(0.003)	(0.081)	(0.004)
	B	1.363 a	0.087 a	5.147 b	0.377 b
		(0.084)	(0.004)	(0.272)	(0.021)
	C	1.012 b	0.070 b	4.826 b	0.360 b
		(0.079)	(0.001)	(0.176)	(0.013)
	D	1.265 a	0.074 b	5.900 a	0.451 a
		(0.083)	(0.002)	(0.226)	(0.012)

Means values in the same column with different letters for each consistency level are significantly different (*p* ≤ 0.05) according to Fisher’s least significant difference. For each mean, standard deviation is in parenthesis.

**Table 6 gels-08-00430-t006:** Registered forces (in N) at different speeds for each thickener at the nectar level for the extrusion and penetration tests.

Thickener	Test Speed (mm s^−1^)	Extrusion with Disc			Extrusion with Cone		Penetration with Sphere	Penetration with Cylinder
		Initial Force	Final Force	Cohesivity	Max. Positive Force	Max. Negative Force	Final Force	Final Force
A	1	0.107 c (0.003)	0.128 c (0.004)	0.087 ab (0.002)	0.337 b (0.014)	0.193 ab (0.005)	0.020 a (0.003)	0.126 a (0.005)
A	3	0.118 b (0.001)	0.142 ab (0.005)	0.092 a (0.009)	0.407 a (0.009)	0.202 ab (0.011)	0.020 a (0.003)	0.130 a (0.004)
A	5	0.119 b (0.001)	0.134 bc (0.006)	0.079 b (0.009)	0.353 b (0.008)	0.188 b (0.012)	0.017 ab (0.003)	0.126 a (0.010)
A	10	0.127 a (0.000)	0.146 a (0.002)	0.083 ab (0.001)	0.423 a (0.014)	0.220 a (0.023)	0.014 b (0.000)	0.134 a (0.006)
B	1	0.095 b (0.001)	0.169 ab (0.015)	0.069 b (0.011)	0.424 a (0.038)	0.163 a (0.113)	0.014 a (0.002)	0.140 a (0.004)
B	3	0.108 a (0.004)	0.163 b (0.014)	0.080 ab (0.009)	0.423 a (0.047)	0.093 a (0.008)	0.017 a (0.005)	0.137 a (0.012)
B	5	0.114 a (0.010)	0.186 a (0.007)	0.095 a (0.008)	0.399 a (0.026)	0.100 a (0.016)	0.016 a (0.003)	0.137 a (0.008)
B	10	0.106 a (0.005)	0.180 ab (0.004)	0.068 b (0.012)	0.362 a (0.026)	0.069 a (0.015)	0.019 a (0.004)	0.138 a (0.006)
C	1	0.093 a (0.009)	0.147 c (0.008)	0.067 a (0.008)	0.364 c (0.022)	0.073 c (0.011)	0.024 a (0.002)	0.143 a (0.005)
C	3	0.091 a (0.005)	0.161 ab (0.006)	0.042 c (0.011)	0.482 b (0.004)	0.109 bc (0.010)	0.015 b (0.002)	0.118 c (0.003)
C	5	0.093 a (0.005)	0.151 bc (0.006)	0.051 bc (0.006)	0.494 b (0.011)	0.139 b (0.018)	0.018 b (0.004)	0.135 b (0.001)
C	10	0.095 a (0.006)	0.164 a (0.005)	0.059 a (0.005)	0.599 a (0.087)	0.204 a (0.050)	0.016 b (0.002)	0.121 c (0.002)
D	1	0.174 d (0.009)	0.195 d (0.006)	0.178 ab (0.015)	0.856 c (0.085)	0.876 a (0.448)	0.024 a (0.002)	0.141 b (0.005)
D	3	0.228 b (0.009)	0.240 b (0.009)	0.170 b (0.013)	1.002 bc (0.072)	0.560 a (0.017)	0.021 ab (0.001)	0.144 b (0.004)
D	5	0.314 a (0.005)	0.321 a (0.003)	0.190 a (0.004)	1.054 b (0.094)	0.586 a (0.053)	0.019 b (0.001)	0.141 b (0.003)
D	10	0.213 c (0.005)	0.217 c (0.003)	0.128 c (0.003)	1.278 a (0.006)	0.587 a (0.026)	0.022 ab (0.001)	0.152 a (0.003)

For the same thickener, the means in the same column with different letters are significantly different (*p* ≤ 0.05) according to Fisher’s least significant difference.

**Table 7 gels-08-00430-t007:** Registered forces (in N) at different speeds for each thickener at pudding level for extrusion and penetration tests.

Thickener	Test Speed (mm s^−1^)	Extrusion with Disc		Extrusion with Cone		Penetration with Sphere	Penetration with Cylinder
		Final Force	Cohesivity	Max. Positive Force	Max. Negative Force	Final Force	Final Force
A	1	0.248 c (0.007)	0.177 a (0.000)	0.594 c (0.028)	0.353 a (0.016)	0.025 a (0.003)	0.186 b (0.004)
A	3	0.277 bc (0.007)	0.168 b (0.003)	0.689 b (0.031)	0.303 b (0.015)	0.026 a (0.002)	0.205 a (0.002)
A	5	0.295 b (0.020)	0.165 a (0.006)	0.731 ab (0.022)	0.307 b (0.015)	0.025 a (0.002)	0.213 a (0.001)
A	10	0.343 a (0.022)	0.163 a (0.003)	0.765 a (0.006)	0.239 c (0.006)	0.027 a (0.002)	0.215 a (0.011)
B	1	0.788 d (0.028)	0.970 ab (0.013)	1.891 b (0.057)	1.603 a (0.021)	0.087 c (0.004)	0.377 c (0.021)
B	3	1.005 c (0.023)	0.996 a (0.026)	1.995 ab (0.097)	1.291 b (0.020)	0.093 bc (0.003)	0.472 b (0.020)
B	5	1.058 b (0.014)	0.946 ab (0.052)	2.052 a (0.068)	1.330 b (0.053)	0.105 ab (0.003)	0.505 b (0.030)
B	10	1.191 a (0.025)	0.929 b (0.010)	1.992 ab (0.099)	1.009 c (0.047)	0.116 a (0.002)	0.546 a (0.012)
C	1	0.759 d (0.016)	0.929 a (0.024)	1.563 c (0.042)	1.308 c (0.024)	0.070 a (0.000)	0.360 d (0.013)
C	3	0.965 c (0.035)	0.910 a (0.010)	2.060 b (0.061)	1.519 ab (0.070)	0.085 a (0.002)	0.437 c (0.004)
C	5	1.014 b (0.007)	0.910 a (0.019)	2.174 b (0.122)	1.582 a (0.102)	0.099 a (0.001)	0.456 b (0.011)
C	10	1.138 a (0.019)	0.901 a (0.028)	2.463 a (0.041)	1.409 bc (0.010)	0.068 a (0.007)	0.508 a (0.005)
D	1	1.088 d (0.023)	1.249 a (0.019)	3.138 b (0.132)	2.774 a (0.071)	0.074 d (0.002)	0.451 d (0.012)
D	3	1.368 c (0.015)	1.200 bc (0.007)	3.287 b (0.036)	2.358 b (0.037)	0.094 c (0.003)	0.601 c (0.022)
D	5	1.506 b (0.063)	1.224 ab (0.024)	3.867 a (0.174)	2.804 a (0.136)	0.103 b (0.003)	0.636 b (0.022)
D	10	1.758 a (0.020)	1.188 c (0.006)	3.886 a (0.180)	2.251 b (0.107)	0.123 a (0.004)	0.767 a (0.015)

For the same thickener, the means in the same column with different letters are significantly different (*p* ≤ 0.05) according to Fisher’s least significant difference.

**Table 8 gels-08-00430-t008:** Mean values of apparent viscosity for the four thickeners at nectar and pudding levels at a shear rate of 10, 50, and 300 s^−1^.

	Nectar			Pudding		
Thickener	ɳ_10_	ɳ_50_	ɳ_300_	ɳ_10_	ɳ_50_	ɳ_300_
A	0.604 a	0.148 a	0.041 b	1.962 c	0.414 d	0.094 c
B	0.504 ab	0.122 b	0.030 c	11.890 a	2.660 b	0.634 b
C	0.355 b	0.105 b	0.027 c	10.408 b	2.375 c	0.582 b
D	0.349 b	0.129 ab	0.055 a	11.933 a	3.324 a	0.792 a

Samples in the same column with different letters indicate a significant difference at *p* < 0.05 according to Fisher’s test.

## Data Availability

Not applicable.

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
