# Peer review of "Characterization of Dysphagia Thickeners Using Texture Analysis—What Information Can Be Useful?"

_gels, 2022, doi:10.3390/gels8070430_

Round 1

Reviewer 1 Report

The article “Characterization of dysphagia thickeners using texture analysis, what information can be useful? Is well written and can be accepted after revising the following comments.

Line 31 to 32: add the names of FDA approved thickeners

There are many researches have been published on this content, what is novel in this?

Does this study have some practical significance for future studies?

The objectives of study need to be revised in the clear version

Why these specific thickeners were chosen, there could be more option with same characteristics.

Need to elaborate the material and method section, Authors should more information regarding the equipment’s parameters.

Section 3.1.2 and 3.2: Need to briefly discuss the results and more literature discussion is required.

Authors should add the limitation of this study

Meanwhile, how it could be effective and pushed industries to use the thickeners with better characteristics.

Need to improve the conclusion part

Author Response

Review 1

The article “Characterization of dysphagia thickeners using texture analysis, what information can be useful?” Is well written and can be accepted after revising the following comments.

Thank you for taking the time in the revision

Line 31 to 32: add the names of FDA approved thickeners

Following reviewer’s suggestion , the name of the different thickeners used in dysphagia have been incorporated into the text and are starch, modified starch, xanthan gum, tara gum, guar gum, and maltodextrins (see Line 32 and 33). The authors hope that this is what the reviewer meant, as we were unsure of the specific question of the reviewer, as there are many thickeners that as additives are approved by FDA and/or EFSA (European Agency). The thickeners used in dysphagia. We hope to have answer to the reviewer

There are many researches have been published on this content, what is novel in this?

The reviewer is right, there are many previous research works that have been focused on the study of dysphagia thickeners. The novelty of this work lies that 1) the use of a texture analyzer for dysphagia thickeners characterization is scarce, this could provide more information about the mechanical properties of dysphagia thickeners,  2) there is no previous exploration of the different tests that one could perform using the texture analyzer, and 3) there is no studies in dysphagia thickeners that link the information obtained by a texture analyzer with the information obtained by studying the shear viscosity.

Following the reviewer’s question and to clarify the novelty of this work to the readers, the following information has been added to the text (Line 93-98):

“In summary, beyond viscosity, there are other mechanical properties of dysphagia thickeners that might help in the patient’s safety and so far, there is no an objective measurement to quantify it. For that, there is a need to explore which texture analyzer tests and the specific settings that could provide more information on dysphagia thickeners’ mechanical properties and if this information complements or overlap the one offered by the information obtained when measuring the shear viscosity..”

Does this study have some practical significance for future studies?

Thank you very much for the question. This study has practical significance for the future. This is because it sets the basic protocol regarding texture analyzer test settings and parameters obtained

The objectives of the study need to be revised in the clear version

Following the reviewer’s suggestion, the authors have written again the objective of the study (see line 99-105):

This study has three objectives. The first one is to explore the information that different texture analyzer tests (type of test, probe, and settings) can provide for dysphagia thickeners’ characterization considering the viscosity levels described in the user instructions of commercial thickeners (nectar and pudding consistencies). The second one is to elucidate if the information provided by the texture analyzer complements or overlaps the information provided by the shear viscosity. And finally, the third objective is to provide a guide with the optimum protocol test to characterize dysphagia thickeners.

Why these specific thickeners were chosen, there could be more option with same characteristics.

The four thickeners of this study were selected as they contain different ingredients (gums, starch and modified starch) and were characterized deeply (viscosity, time dependence and effect in the different matrices) in a previous study that we published recently (Badia‐Olmos, C., Laguna, L.*, Rizo, A., & Tárrega, A. (2022). Dysphagia thickeners in context of use. Changes in thickened drinks’ viscosity and thixotropy with temperature and time of consumption. Journal of texture studies).

The authors have added a table with the whole composition andthe following information to the manuscript (see Line 111-113):

“Selected thickeners varied in composition and represented the common ingredients used by dysphagia patients”

Need to elaborate the material and method section, Authors should more information regarding the equipment’s parameters.

Thank you for the comment. More information about the Texture analyzer equipment has been added at the beginning of 2.3. section (see Line 131-135). Along with the different subsections (2.3.1 and 2.3.2) all the parameters for each test had been defined. 

Section 3.1.2 and 3.2: Need to briefly discuss the results and more literature discussion is required.

Thanks for the suggestion, more information has been added in both sections. Please, see Lines 235 to 297

Authors should add the limitation of this study

Thank you, the following limitation has been added (see Line 354-359)

Meanwhile, how it could be effective and pushed industries to use the thickeners with better characteristics.

Thank you very much for the thought. This study aimed to set a methodology to characterize dysphagia thickeners. In future studies, thickeners in real conditions (with participants with dysphagia) will be assessed by videofluoroscopy. Authors hope that with that kind of information, they will be able to propose improvements to the dysphagia thickeners industry

Need to improve the conclusion part

More information has been added to the conclusion part (see lines 392-399)

Reviewer 2 Report

This work aims to explore different texture analyzer settings (type of test, probe, and protocol) for four commercial dysphagia thickeners at the shear viscosity levels recommended by manufacturers. This kind of studies are relevant when studying safe food matrices for swallowing in people suffering from oropharyngeal dysphagia.

This study besides rather standard and often obtained results, contains some new approaches. In my opinion, only a much more profound study would justify the publication of this work.

Some comments comments that deserve further consideration are described below:

Introduction

- This work must be complemented with critical references in this research field. Several efforts have been put into enriching this topic in the last decade.

- It is neccesary to describe advantages and drawbacks of proposing the use of texture analyzer tests in a medical environment focused ondysphagia thickeners’ characterization.

- Line 34-35: References must be included to support this statement. (This problem is found throughout the text).

Materials and methods

- A table describing the compositional characteristics of each commercial thickener used (n=4) must be provided.

- This study proposed to use a texture analyzer to characterize different commercial dysphagia thickeners when dissolved in water. ¿Is it representative to use water as the working fluid?

- Lines 101-102: Include references to support this statement: Four tests frequently used in the evaluation of textural attributes of semi-solid food products were performed: two back-extrusion tests and two penetration tests…

- Lines 104-105:

1) Why these experimental assays were conducted at 1, 3, 5, and 10 mm/s? Criteria used for this choice are required. This is mandatory for detailed information on texture analysis , independently that in section 3.2 the authors mention: Regarding the selection of the speed to perform the test, the presented results show that the information provided by any speed reflects the same information trend, and one speed would be enough to understand the textural behavior of the samples. For ease of results interpretation, the selection of 1 mm s-1 could be used.

2) Why the texture analyzer tests were performed at 25 °C? Is this value of temperature representative of the human phisiology when swallowing a food matrix?

- Section 2.4. How do the authors explain that the thickener flow curves were performed at 37 °C (normal body temperature) instead of 25 °C?

Results (and discussion?)

The discussion related to the results obtained from the extrusion tests is weakly developed and poorly understood. In addition, how do the authors rationalize the differences found among thickeners based on compositional data analysis? Same situation is observed in sections 3.1.2. (Penetration test), 3.2 (effect of test speed) and 3.3. (Relationship of texture analyzer parameters with traditional viscosity measurements). This study lacks a broader theoretical discussion to avoid being confined to a descriptive mode of presentation or discussion.

Author Response

Review 2

This work aims to explore different texture analyzer settings (type of test, probe, and protocol) for four commercial dysphagia thickeners at the shear viscosity levels recommended by manufacturers. This kind of studies are relevant when studying safe food matrices for swallowing in people suffering from oropharyngeal dysphagia.

This study besides rather standard and often obtained results, contains some new approaches. In my opinion, only a much more profound study would justify the publication of this work.

Authors thank the reviewer for the time invested in improved the paper.

Some comments comments that deserve further consideration are described below:

Introduction

- This work must be complemented with critical references in this research field. Several efforts have been put into enriching this topic in the last decade.

Authors have increased the state of the art by remarking on other areas of food and liquids improvement for dysphagia. Also, the latest physical characterization of dysphagia thickeners (tribology, extensional rheology) has been also added.

Please, see lines 53 to 64.

References added:

  1. Wang, B. et al. Rheological properties of waxy maize starch and xanthan gum mixtures in the presence of sucrose. Carbohydr. Polym. 77, 472–481 (2009).
  2. Leonard, R. J., White, C., McKenzie, S. & Belafsky, P. C. Effects of bolus rheology on aspiration in patients with dysphagia. J. Acad. Nutr. Diet. 114, 590–594 (2014).
  3. Hadde, E. K. & Chen, J. Shear and extensional rheological characterization of thickened fluid for dysphagia management. J. Food Eng. 245, 18–23 (2019).
  4. Moret-Tatay, A., Rodríguez-García, J., Martí-Bonmatí, E., Hernando, I. & Hernández, M. J. Commercial thickeners used by patients with dysphagia: Rheological and structural behaviour in different food matrices. Food Hydrocoll. 51, 318–326 (2015).
  5. Kohyama, K. et al. Electromyographic texture characterization of hydrocolloid gels as model foods with varying mastication and swallowing difficulties. Food Hydrocoll. 43, 146–152 (2015).
  6. Ross, A. I. V, Tyler, P., Borgognone, M. G. & Eriksen, B. M. Relationships between shear rheology and sensory attributes of hydrocolloid-thickened fluids designed to compensate for impairments in oral manipulation and swallowing. J. Food Eng. 263, 123–131 (2019).
  7. Waqas, M. Q., Wiklund, J., Altskär, A., Ekberg, O., & Stading, M. (2017). Shear and extensional rheology of commercial thickeners used for dysphagia management. Journal of texture studies, 48(6), 507-517.
  8. Vieira, J. M., Oliveira Jr, F. D., Salvaro, D. B., Maffezzolli, G. P., de Mello, J. B., Vicente, A. A., & Cunha, R. L. (2020). Rheology and soft tribology of thickened dispersions aiming the development of oropharyngeal dysphagia-oriented products. Current research in food science, 3, 19-29.
  9. Torres, O., Yamada, A., Rigby, N. M., Hanawa, T., Kawano, Y., & Sarkar, A. (2019). Gellan gum: A new member in the dysphagia thickener family. Biotribology, 17, 8-18.

- It is necessary to describe advantages and drawbacks of proposing the use of texture analyzer tests in a medical environment focused ondysphagia thickeners’ characterization.

Thanks for the suggestion, authors have added the following information to the end of the manuscript (Please, see Lines 376-687)

- Line 34-35: References must be included to support this statement. (This problem is found throughout the text).

Thank you, reference (Cichero 2013a) has been added. Please, see Line 36

Materials and methods

- A table describing the compositional characteristics of each commercial thickener used (n=4) must be provided.

Thanks for the suggestion, a compositional table has been added to the reviewed manuscript version.

- This study proposed to use a texture analyzer to characterize different commercial dysphagia thickeners when dissolved in water. ¿Is it representative to use water as the working fluid?

It is, as in a previous paper recently published by the authors, we studied the effect of others matrices in the thickeners

Badia‐Olmos, C., Laguna, L.*, Rizo, A., & Tárrega, A. (2022). Dysphagia thickeners in context of use. Changes in thickened drinks’ viscosity and thixotropy with temperature and time of consumption. Journal of texture studies

- Lines 101-102: Include references to support this statement: Four tests frequently used in the evaluation of textural attributes of semi-solid food products were performed: two back-extrusion tests and two penetration tests…

Bibliography has been added:

Liu, Y. X., Cao, M. J., & Liu, G. M. (2019). Texture analyzers for food quality evaluation. In Evaluation Technologies for Food Quality (pp. 441-463). Woodhead Publishing.

- Lines 104-105:

1) Why these experimental assays were conducted at 1, 3, 5, and 10 mm/s? Criteria used for this choice are required. This is mandatory for detailed information on texture analysis , independently that in section 3.2 the authors mention: Regarding the selection of the speed to perform the test, the presented results show that the information provided by any speed reflects the same information trend, and one speed would be enough to understand the textural behavior of the samples. For ease of results interpretation, the selection of 1 mm s-1 could be used.

During the test, the probe keeps a constant speed. For other products, as crips dry products, to record the noise, the speed is crucial, but for other systems as the thickened drinks was not clear enough and not describe in previous literature how the speed might affect it. Authors could not know the outcome before doing the experiments

2) Why the texture analyzer tests were performed at 25 °C? Is this value of temperature representative of the human phisiology when swallowing a food matrix?

- Section 2.4. How do the authors explain that the thickener flow curves were performed at 37 °C (normal body temperature) instead of 25 °C?

Thank you for rising this very interesting point. In a previous work, we did measure this same four thickeners at 25°C (Badia‐Olmos, C., Laguna, L.*, Rizo, A., & Tárrega, A. 2022. Dysphagia thickeners in context of use. Changes in thickened drinks’ viscosity and thixotropy with temperature and time of consumption. Journal of texture studies)  and authors thought that it will be more realistic to perform the rheology at 37C, however, we could not do that with the texture analyzer as we did not have a peltier plate or camera coupled in the texture analyzer that allowed to maintain that temperature. This has been explained as a limitation of this study (see Line 370-374)

Results (and discussion?)

The discussion related to the results obtained from the extrusion tests is weakly developed and poorly understood. In addition, how do the authors rationalize the differences found among thickeners based on compositional data analysis? Same situation is observed in sections 3.1.2. (Penetration test), 3.2 (effect of test speed) and 3.3. (Relationship of texture analyzer parameters with traditional viscosity measurements). This study lacks a broader theoretical discussion to avoid being confined to a descriptive mode of presentation or discussion.

Thanks for your point of view. Authors have added more discussion to the results and discussion section (please see Lines 235-297). Also, authors wanted to emphasized that this paper aimed to study if and how the texture analyzer can be used to characterize dysphagia thickeners. It did not aim to find difference among thickeners or explain why the behave different. Taking into considerations ‘reviewers’ opinion, the objective of this study has also been rewritten (please see Lines 99-105)

Round 2

Reviewer 1 Report

All the comments are well addressed by reviewers.

Author Response

Thank you

Reviewer 2 Report

The new version of this work must improve the following issues:

1. Table 1. Include percentages for each ingredient included. This information is critical to identify each thickener tested and address correct discussions.

2. Section 2.2. Preparation of the thickened drinks. The authors must specify that water is a representative fluid for this purpose and mention other liquids fluids (e.g., coffee, orange juice, and milk) as reported by them in https://doi.org/10.1111/jtxs.12685.

3. Section 2.3 Texture analyzer tests. In this section, the authors must explain why the assays were performed at 25°C and the warning of using this information if it is known that the normal body temperature is 37 °C. This point is relevant when applying and analyzing results obtained from the thickener flow curves. This new explanatory text will complement the text in red (lines 370-374) included in section 3.3 about the limitations of this study.

Author Response

The new version of this work must improve the following issues:

  1. Table 1. Include percentages for each ingredient included. This information is critical to identify each thickener tested and address correct discussions.

Thank you very much for the suggestion, however as samples are commercial thickeners, we do not have that information.

  1. Section 2.2. Preparation of the thickened drinks. The authors must specify that water is a representative fluid for this purpose and mention other liquids fluids (e.g., coffee, orange juice, and milk) as reported by them in https://doi.org/10.1111/jtxs.12685.

Thank you very much for the suggestion.

As we include now in this version of the manuscript, water is the most used drink to avoid dehydration and to calm thirstiness. Other drinks can be used either hot (coffee, tea, and milk) or cold (juice, soda, and milk). In general, in previous work, we demonstrate that as the temperature increased (from 10 to 50C) there was an increment in viscosity, probably due to greater starch gelatinization. Matrix also had an influence, i.e., milk and juice that have soluble solid had more viscosity than coffee.

This information has been added see lines 119-124

  1. Section 2.3 Texture analyzer tests. In this section, the authors must explain why the assays were performed at 25°C and the warning of using this information if it is known that the normal body temperature is 37 °C. This point is relevant when applying and analyzing results obtained from the thickener flow curves. This new explanatory text will complement the text in red (lines 370-374) included in section 3.3 about the limitations of this study.

Following the suggestion of the reviewer the following test was added in lines 144 to 147

“All tests were conducted at 1, 3, 5, and 10 mm/s, and at 25 °C. This was the room temperature as no device to maintain the temperature in the texture analyzer was available. The selection of the temperature is important as the physical properties depend on it, for example viscosity, according to Arrhenius equation, depends on temperature.”